# Simple Unsupervised Object-Centric Learning for Complex and Naturalistic Videos

**Gautam Singh**[*]
Rutgers University
singh.gautam@rutgers.edu

**Yi-Fu Wu**
Rutgers University
yifu.wu@gmail.com

**Sungjin Ahn**
KAIST
sjn.ahn@gmail.com

## Abstract

Unsupervised object-centric learning aims to represent the modular, compositional, and causal structure of a scene as a set of object representations and thereby promises to resolve many critical limitations of traditional single-vector representations such as poor systematic generalization. Although there have been many remarkable advances in recent years, one of the most critical problems in this direction has been that previous methods work only with simple and synthetic scenes but not with complex and naturalistic images or videos. In this paper, we propose STEVE, an unsupervised model for object-centric learning in videos. Our proposed model makes a significant advancement by demonstrating its effectiveness on various complex and naturalistic videos unprecedented in this line of research. Interestingly, this is achieved by neither adding complexity to the model architecture nor introducing a new objective or weak supervision. Rather, it is achieved by a surprisingly simple architecture that uses a transformer-based image decoder conditioned on slots and the learning objective is simply to reconstruct the observation. Our experiment results on various complex and naturalistic videos show significant improvements compared to the previous state-of-the-art. https://sites.google.com/view/slot-transformer-for-videos

## 1 Introduction

The goal of object-centric representation learning is to represent the inherent structure of the physical world such as compositionality and causality as a set of representation vectors (and their relations) each corresponding to a conceptual entity module such as an object (Greff et al., 2020; Schölkopf et al., 2021). Many previous works have demonstrated the potential of this approach as a way of resolving the key limitations of traditional single-vector representations (Greff et al., 2020; Schölkopf et al., 2021; Dittadi et al., 2021; Eslami et al., 2016; Bapst et al., 2019; Mambelli et al., 2022; Ghasemipour et al., 2022). For example, considering object-centric representations as independent mechanisms helps realize systematic and zero-shot generalization for image generation (Singh et al., 2022; Chen et al., 2021). Also, the ability to decompose a visual observation into a set of discrete knowledge modules has shown to be useful for visual reasoning (Zhou et al., 2021; Wu et al., 2021).

One of the most critical challenges remaining in object-centric representation learning is to successfully apply it to complex and natural images or videos. Achieving this has primary importance because it opens up a way to utilize the enormous amounts of images and videos available on the internet and thereby unleash the full potential of this method by applying it at scale — the machine learning community has recently observed the power of scale in large-scale language models (Brown et al., 2020; Bommasani et al., 2021).

---

[*]Correspondence to singh.gautam@rutgers.edu and sjn.ahn@gmail.com.

36th Conference on Neural Information Processing Systems (NeurIPS 2022).

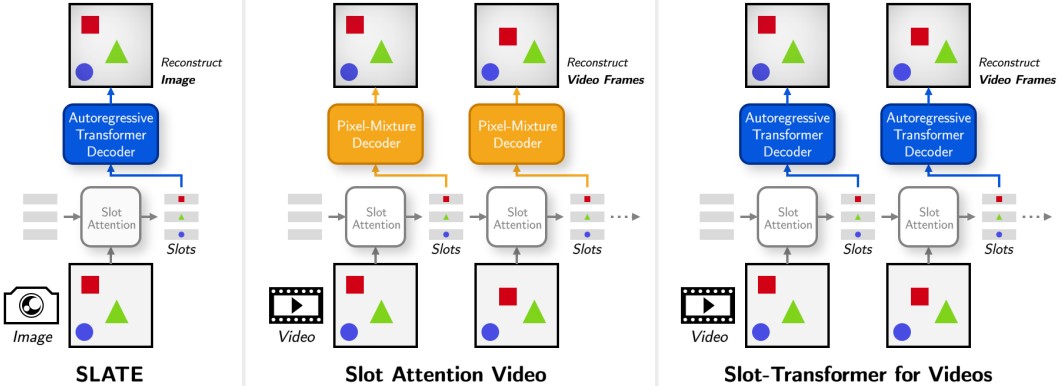

Figure 1: **Overview of the proposed model. Left:** SLATE auto-encoder combines the representational bottleneck of slots with an auto-regressive transformer decoder and shows unsupervised object discovery in visually complex images. However, the question of whether this framework can deal with complex and naturalistic videos is unexplored. **Middle:** Conventional object-centric video models such as Slot Attention Video deal with videos by applying slot attention recurrently on the frames and applying a pixel-mixture decoder to reconstruct the frames. However, in the fully unsupervised setting, these lack the ability to handle complex and naturalistic videos. **Right:** Our model *Slot Transformer for Videos* provides a simple and minimal architecture leveraging an auto-regressive transformer decoder that can effectively handle complex and naturalistic videos without any supervision.

However, despite a significant amount of effort and many remarkable achievements in recent years, we do not yet have an unsupervised object-centric learning method that works well on complex and naturalistic videos. In fact, when considering the high complexity of natural images, the ultimate goal of making the concept of objects emerge without any supervision, i.e., only by observing, indeed remains quite elusive. As such, previous works have been shown to work only on toy or synthetic images/videos or resorted to utilizing some supervision such as optical flow or annotation on the initial frame (Kipf et al., 2021).

In this paper, we propose an unsupervised model for object-centric learning in videos, called STEVE (Slot-TransformEr for VidEos). Our proposed model makes a significant advancement by demonstrating its effectiveness for the first time on various complex and naturalistic videos unprecedented in this line of research. Interestingly, this is achieved by neither adding much complexity to the model architecture nor by introducing a new objective or weak supervision. Rather, it is achieved by a surprisingly simple architecture, comparable to or simpler than most of the previous methods that have not worked well on complex videos. Moreover, the learning objective is just to reconstruct. We believe this simplicity is one of the best strengths of our model.

Behind this success is the adoption of the transformer-based slot decoder recently proposed in SLATE (Singh et al., 2022). Because it is demonstrated in the paper that this decoder is the key to dealing with some visually complex synthetic images, it is of deep interest to the community (i) *whether and how* it can be extended to a temporal model utilizing dynamical observations, (ii) if it can eventually deal with complex naturalistic videos, and (iii) what lessons we can learn from this. To focus on investigating these central questions systematically, we intentionally choose not to pursue architectural novelty at the expense of adding complexity but to test a minimal architecture that combines the SLATE decoder with a standard slot-level recurrence model. Our experiment results on various complex and naturalistic videos show that the proposed architecture significantly outperforms previous state-of-the-art baseline models.

From these experiments, we discover important new knowledge to share with the community. We find that SLATE can deal with complex naturalistic images reasonably well despite not using temporal information. Although as a static model, SLATE has a fundamental limitation that it cannot take advantage of temporal information, e.g., maintaining consistent object identity across time, if we only look at its frame-level segmentation ability, it already performs better than the previous state-of-the-art temporal model that applies slot attention to videos (Kipf et al., 2021). Crucially, we also find that SLATE alone is not enough since our model, STEVE, shows much better frame segmentation

than SLATE, while also providing consistent tracking of slots in videos, which SLATE cannot do. In particular, we find that on some complex videos such as MOVi-Tex, SLATE fails significantly, suggesting that learning jointly from both temporal information and the powerful SLATE decoder is significant and essential to realize the full potential of object-centric representation learning in videos. Furthermore, our analysis shows that our model is robust to challenges such as static objects and camera motion and that it also generalizes well to out-of-distribution number of objects and unseen textures. Lastly, we propose several new datasets that are significantly more complex than those tackled by the previous unsupervised models.

## 2  Preliminaries

In unsupervised object-centric representation learning, the goal is to learn to obtain the representation of an observation as a set of object vectors, referred to as *slots* (Locatello et al., 2020). A common approach for this is to auto-encode: a slot-encoder $f_\phi^{\text{enc}}(\mathbf{x})$ extracts a set of slot representations $\mathbf{s} = \{\mathbf{s}_1, \ldots \mathbf{s}_N\}$ from an image $\mathbf{x}$ and then a decoder $f_\phi^{\text{dec}}(\mathbf{s})$ makes a reconstruction $\hat{\mathbf{x}}$ from these slots, where $\mathbf{x} \in \mathbb{R}^{H \times W \times C}$ and $\mathbf{s}_n \in \mathbb{R}^D$. In the encoder, the slots compete with each other to learn about which area of the input will be explained by which slots. This provides a form of information bottleneck encouraging a slot to attend an area corresponding to a meaningful conceptual entity like an object. During training, a reconstruction objective provides the learning signal for training the complete model. One nice property of this method is that it only requires reconstructing the image and does not require any weak supervision or object-aware auxiliary loss terms. Ideally, we want to maintain this simplicity in video models, but current state-of-the-art requires weak supervision and optical flow to make it work on complex naturalistic videos (Kipf et al., 2021).

**Mixture-based Decoder.** The emergence of object-centric slots is strongly dependent on what decoder is used. The traditional way that most of the previous methods have taken is the mixture-based decoder or some variant of it. In this approach, the decoder decodes each slot $\mathbf{s}_n$ to obtain an object image $\hat{\mathbf{x}}_n$ and an alpha mask $\mathbf{m}_n$ via decoding functions $g_\theta^{\text{RGB}}$ and $g_\theta^{\text{mask}}$, respectively. The decoded object images are then weighted-summed to obtain the full image $\hat{\mathbf{x}}$:

$$\hat{\mathbf{x}}_n = g_\theta^{\text{RGB}}(\mathbf{s}_n), \qquad \mathbf{m}_n = \frac{\exp g_\theta^{\text{mask}}(\mathbf{s}_n)}{\sum_{m=1}^N \exp g_\theta^{\text{mask}}(\mathbf{s}_m)}, \qquad \hat{\mathbf{x}} = \sum_{n=1}^N \mathbf{m}_n \cdot \hat{\mathbf{x}}_n.$$

These decoder networks are implemented using a CNN. The reconstruction objective is taken to be the squared error between the input and the reconstructed image i.e. $\mathcal{L}_{\text{mixture}} = \|\hat{\mathbf{x}} - \mathbf{x}\|^2$. A critical limitation of this approach is that it has never been successful in dealing with scenes with high visual complexity like natural images.

**Autoregressive Slot-Transformer Decoder.** Recently, Singh et al. (2022) have challenged this traditional perspective that we need a mixture decoder for the emergence of objectness in slots. Arguing that a mixture decoder severely limits the interaction among slots and the reconstruction quality—hence a problem in obtaining good learning signal for complex images—they proposed a new architecture, SLATE, using a powerful autoregressive decoder based on a transformer conditioned on the slots (Chen et al., 2020). However, many critical questions about this new approach still remain because the focus of the SLATE paper is on image generation ability. To get a deeper understanding of the object representations that a model like SLATE produces, we would need to thoroughly investigate the quantitative evidence about its scene decomposition ability, its ability to deal with visual complexity close to that of natural images, and finally, the effect of extending this architecture to a temporal model. We aim to answer these in this paper.

## 3  STEVE: Slot Transformer for Videos

To study the effect of extending the slot-transformer decoder to a temporal model that can deal with videos, we choose to propose a minimal architecture. Specifically, our proposed model, STEVE, combines three main components: (1) a CNN-based image encoder, (2) a recurrent slot encoder that updates slots temporally with recurrent neural networks (RNNs), and (3) the slot-transformer decoder of SLATE. Although we may obtain additional performance gains by exploring further architectural novelty at the cost of additional complexity, we choose to investigate this minimal architecture to focus on our goal of investigating slot-transformer decoder models for video.

Given a video consisting of frames $\mathbf{x}_1, \ldots \mathbf{x}_T$, we want to maintain $N$ slots $\mathbf{s}_t = \{\mathbf{s}_{t,1}, \ldots, \mathbf{s}_{t,N}\}$ for each time-step $t \in \{1, \ldots T\}$ by starting the temporal update from initial slots $\mathbf{s}_0$. We apply a recurrent slot encoder $f_\phi^{\text{slot-rnn}}$ at each step $t$ to update the slot representations from $\mathbf{s}_{t-1}$ to $\mathbf{s}_t$ using the information from the input frame $\mathbf{x}_t$.

$$\mathbf{s}_t = f_\phi^{\text{slot-rnn}}(\mathbf{x}_t; \mathbf{s}_{t-1}), \qquad\qquad \mathbf{s}_0 = \texttt{initialize}().$$

It is expected that each slot $n$ would represent one object and track it consistently over time in a given video. For each frame, the slot representation $\mathbf{s}_t$ is used to reconstruct the input frame $\mathbf{x}_t$ using the slot-transformer decoder. For doing such reconstruction, each frame $\mathbf{x}_t$ is treated as a sequence of discrete tokens provided by a discrete VAE encoder. Given the slots $\mathbf{s}_t$, the slot-transformer decoder learns to predict this sequence of tokens auto-regressively by minimizing a cross-entropy loss.

$$\mathcal{L}_{\text{CE}} = \sum_{t=1}^{T}\sum_{l=1}^{L} \text{CE}(\mathbf{z}_{t,l}, \mathbf{o}_{t,l}), \quad \mathbf{z}_{t,1}, \ldots, \mathbf{z}_{t,L} = f_\phi^{\text{dVAE}}(\mathbf{x}_t), \quad \mathbf{o}_{t,l} = g_\theta^{\text{slot-transformer}}(\mathbf{s}_t; \mathbf{z}_{t,1} \ldots \mathbf{z}_{t,l-1}).$$

where $\text{CE}(\cdot, \cdot)$ denotes cross-entropy loss, $\mathbf{z}_t = \{\mathbf{z}_{t,1}, \ldots, \mathbf{z}_{t,L}\}$ are $L$ discrete tokens that act as prediction targets for the transformer at time $t$ and $\mathbf{o}_{t,l}$ contains the log-probabilities predicted by the transformer for the token at position $l$ for frame $t$. To train the discrete VAE, an image reconstruction loss $\mathcal{L}_{\text{dVAE}}$ is used as in SLATE.

$$\mathcal{L}_{\text{dVAE}} = \sum_{t=1}^{T} \|\hat{\mathbf{x}}_t - \mathbf{x}_t\|_2^2, \qquad \mathbf{z}_{t,1}, \ldots \mathbf{z}_{t,L} = f_\phi^{\text{dVAE}}(\mathbf{x}_t), \qquad \hat{\mathbf{x}}_t = g_\theta^{\text{dVAE}}(\mathbf{z}_{t,1}, \ldots \mathbf{z}_{t,L}).$$

The complete model is trained using $\mathcal{L}_{\text{STEVE}} = \mathcal{L}_{\text{CE}} + \mathcal{L}_{\text{dVAE}}$.

**Recurrent Slot Encoder.** Several recent works have implemented recurrent object-centric models by incorporating an RNN-based dynamics model to update slots over timesteps (Veerapaneni et al., 2019; Jiang et al., 2020; Kossen et al., 2020; Lin et al., 2020a; Stanić & Schmidhuber, 2019). For our model, we leverage a slot-based recurrent encoder that is also used as the backbone of Slot Attention Video (Kipf et al., 2021). Our slot-based recurrent encoder works as follows. At the start of an episode, we provide initial slots $\mathbf{s}_0$ by randomly sampling from a Gaussian distribution with learned mean and variance. For each input frame $\mathbf{x}_t$, we compute an encoding in the form of a feature map using a backbone CNN. This feature map is flattened to obtain a set of input features $\mathbf{e}_t = \{\mathbf{e}_{t,1}, \mathbf{e}_{t,2}, \ldots, \mathbf{e}_{t,HW}\}$. Next, the slots $\mathbf{s}_{t-1}$ perform attention on the features $\mathbf{e}_t$ and use the attention result to update the slots to $\mathbf{s}_t$. For this, the slots $\mathbf{s}_{t-1}$ compute attention weights $\mathcal{A}_t = \{\mathcal{A}_{t,1}, \ldots, \mathcal{A}_{t,N}\}$ over the input features, where $\mathcal{A}_{t,n}$ are the attention weights for slot $n$. The attention weights are used to perform an attention-weighted sum of the input features. The attention result $\mathbf{r}_{t,n}$ is then used to update the respective slot using an RNN $f_\phi^{\text{RNN}}$.

$$\tilde{\mathbf{s}}_{t,n} = f_\phi^{\text{RNN}}(\mathbf{r}_{t,n}, \mathbf{s}_{t-1,n}), \quad \mathbf{r}_{t,n} = \frac{\sum_{i=1}^{HW} \mathcal{A}_{t,n,i} \cdot v(\mathbf{e}_{t,i})}{\sum_{j=1}^{HW} \mathcal{A}_{t,n,j}}, \quad \mathcal{A}_t = \operatorname*{softmax}_N \left( \frac{q(\mathbf{s}_{t-1}) \cdot k(\mathbf{e}_t)^T}{\sqrt{D}} \right).$$

Here, $q$, $k$ and $v$ are linear projections and $D$ is the output size of the projections. Finally, the slots are made to interact via an interaction network: $\mathbf{s}_t = f_\phi^{\text{interact}}(\tilde{\mathbf{s}}_t)$. Following Kipf et al. (2021), we use the slots $\tilde{\mathbf{s}}_t$ before the interaction step for reconstruction. For complete details of the architecture, see Appendix B.

## 4 Related Work

**Unsupervised Object-Centric Representation Learning in Images.** Unsupervised object-centric learning methods for static scenes have received significant interest in recent years (Greff et al., 2016; Burgess et al., 2019; Greff et al., 2019; Locatello et al., 2020; Greff et al., 2017; Engelcke et al., 2020, 2021; Eslami et al., 2016; Crawford & Pineau, 2019b; Lin et al., 2020b; Jiang & Ahn, 2020; Deng et al., 2020; Chen et al., 2021; Anciukevicius et al., 2020; von Kügelgen et al., 2020; Greff et al., 2020). These models learn through reconstruction, commonly adopting a mixture-based decoder. Another line of work has pursued object discovery by minimizing the mutual information between the predicted segments (Savarese et al., 2021; Yang et al., 2020). Approaches based on self-supervised representation learning have also been investigated (Caron et al., 2021; Löwe et al., 2020; Wang et al., 2022). Recently, scene decomposition ability has been shown to emerge in energy-based models (Du

et al., 2021a; Yu et al., 2021). Complex-valued neural networks have also shown such emergence Lowe et al. (2022). Closely related to our work, Lamb et al. (2021) and Singh et al. (2022) have shown that a slot-based encoder combined with a transformer decoder can enable object emergence in visually complex images. However, all of these approaches deal only with static images unlike ours which can be applied to videos.

**Unsupervised Object-Centric Representation Learning in Videos.** A large body of work has approached fully-unsupervised video segmentation and tracking by combining recurrent slot-based encoders with a reconstruction objective. In this class, (Kosiorek et al., 2018; Stanić & Schmidhuber, 2019; Jiang et al., 2020; Crawford & Pineau, 2019a; Lin et al., 2020a; Wu et al., 2021; Singh et al., 2021; He et al., 2019) use bounding boxes for tracking. A parallel line of research learns to localize objects via segmentation masks (Greff et al., 2017; Van Steenkiste et al., 2018; Veerapaneni et al., 2019; Watters et al., 2019; Weis et al., 2020; Du et al., 2020; Kipf et al., 2021; Kabra et al., 2021; Zoran et al., 2021; Besbinar & Frossard, 2021; Creswell et al., 2020, 2021). Our work lies along this line of research. Other approaches have considered using contrastive losses (Kipf et al., 2019; Carvalho et al., 2020) while some works have explored unsupervised object-centric learning in 3D scenes (Chen et al., 2021; Henderson & Lampert, 2020; Crawford & Pineau, 2020; Stelzner et al., 2021; Du et al., 2021b; Kabra et al., 2021). While dealing with 3D scenes would be an important future direction for our work, it is orthogonal to our current focus. However, unlike ours, all of the above works have only been successful on visually simple videos. In Appendix D, we discuss the related work that deals with unsupervised video object segmentation using motion cues and unsupervised segmentation propagation.

## 5   Experiments

**Datasets.** We evaluate our model on 8 datasets. These include 6 procedurally generated datasets: CATER (Girdhar & Ramanan, 2020), CATERTex, MOVi-Solid, MOVi-Tex, MOVi-D, and MOVi-E (Greff et al., 2022); and 2 natural datasets: Traffic and Aquarium. Out of these, 5 datasets are our contributions in this work which we now describe. *i) CATERTex.* We generated it by combining objects and textures of CLEVRTex (Karazija et al., 2021) with the object motion of CATER (Girdhar & Ramanan, 2020). *ii) MOVi-Solid.* We used Kubric (Greff et al., 2022) to generate this dataset and introduced textured backgrounds and more complex shapes to the *MOVi* dataset proposed by Kipf et al. (2021). *iii) MOVi-Tex.* We added the textures from the Describable Textures dataset (Cimpoi et al., 2014) as materials to the MOVi-Solid dataset to make this dataset. By applying the same material on all surfaces, the objects and their borders become significantly harder to distinguish, resulting in much higher visual complexity over MOVi-Solid. *iv) Traffic and Aquarium.* We created these two natural datasets by collecting a 6-hour long video stream from Youtube. For training, all models use only the raw videos as input with no other supervision or input whatsoever. For evaluation, we use the ground truth instance-level masks. For sample video frames and more details, see Appendix A. We will release all the proposed datasets which, we believe, will facilitate future research. For existing benchmarks i.e. CATER, MOVi-D and MOVi-E, we use the standard train and test splits as prescribed by their respective authors. We also note that in the MOVi benchmark i.e. MOVi-A to E, versions D and E are the two most challenging versions. Thus, we consider these sufficient for evaluating our hypothesis.

**Baselines.** We compare our performance with three baselines which are state-of-the-art in unsupervised object-centric scene decomposition. These include two baselines that use mixture-based reconstruction: Slot Attention Video (Kipf et al., 2021) and OP3 (Veerapaneni et al., 2019); and one baseline which uses transformer-based reconstruction: SLATE (Singh et al., 2022). For Slot Attention Video, we use the fully unsupervised version that is trained only using the image reconstruction objective without optical flow or label information in the first frame. For simplicity, we will refer to this unsupervised Slot Attention Video as SAVi. Comparison with SAVi is important because our model is most similar to it with one main difference: our model uses transformer-based decoding while SAVi uses mixture-based decoding. Therefore, a comparison of our model with SAVi will provide a clear insight into the effect of the decoding approach. For the baseline SLATE, we use the same CNN backbone as our model for a fair comparison. In line with the previous works, for SAVi and OP3, we take their decoding masks to be their predicted segmentation. Since our model STEVE and the baseline SLATE are based on transformer decoder, they do not provide decoding masks. Therefore, we use their input attention masks as our predicted segmentation. Note that this

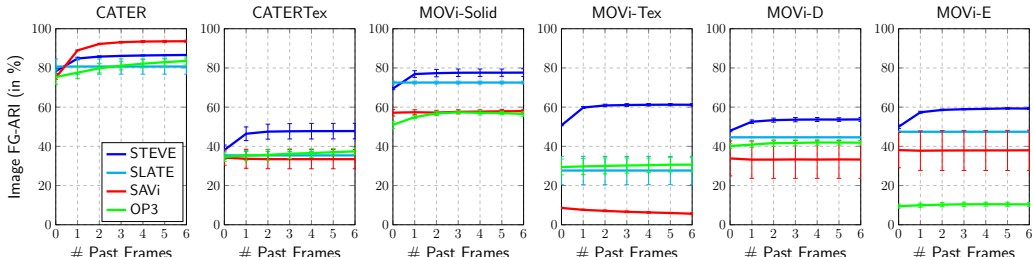

Figure 2: **Unsupervised Image Segmentation.** On the $y$-axis, we plot Image FG-ARI of per-frame segmentation. Along the $x$-axis, we show how the segmentation is affected as more frames are seen in a video. The FG-ARI of SLATE is computed on one-frame 'videos' and is broadcasted along the $x$-axis to facilitate comparison with other models.

difference gives an advantage to the baselines but not to our model as in section 5.3, our analysis will show that the decoding masks perform better than the input attention masks.

## 5.1 Unsupervised Image Segmentation

In this section, we evaluate how effective our model is for unsupervised image segmentation in videos. We report *Image FG-ARI* which measures how well the predicted segmentation matches the ground truth segmentation of a given single image. In line with the previous works (Locatello et al., 2020; Kabra et al., 2021; Kipf et al., 2021), we consider only the foreground segments of the ground truth. In Figure 2, we plot the Image FG-ARI of per-frame segmentation.

**Benefit over Mixture-based Methods.** Comparing STEVE with SAVi and OP3, we see that STEVE achieves significantly higher FG-ARI compared with the baselines in 5 out of 6 datasets. All of these 5 datasets are characterized by complex textures. Therefore, our superior performance shows that our model is more effective in dealing with textured scenes compared to the baselines. Because the main difference of STEVE with SAVi is the reconstruction approach, this result provides strong evidence that the transformer-based reconstruction is the key driver of this improvement. Analyzing the qualitative results in Figures 4 (left), 5 and 9 reveals that the baselines fail completely in CATERTex, MOVi-Tex, MOVi-D, and MOVi-E, most commonly by splitting the image into fixed patch regions instead of meaningful object regions. Unlike simple datasets used in prior works such as CLEVR or CATER that provide strong color contrast between objects, our datasets such as CATERTex and MOVi-Tex provide no such color cues. Despite this, our model can remarkably discover the objects. With the state-of-the-art baselines completely failing, it is for the first time in this line of research that such complex datasets have been effectively handled. In one dataset i.e. CATER, our performance is comparable to OP3 and slightly worse than SAVi. As CATER does not have complex textures, it is not surprising to see baselines performing well on this dataset. In Figure 2, we also find that as the models see more frames, the segmentation tends to improve.

**Benefit of Temporal Learning.** We also analyze whether training STEVE on videos has any benefit in image segmentation compared to SLATE. As SLATE is applicable only for static images, we train and test it on one frame 'videos' of our datasets. Comparing STEVE with SLATE, we note that STEVE performs consistently better in all datasets. Noteworthy is the gap in MOVi-Tex which is especially large. This suggests that training on temporal data i.e. videos can be helpful for learning to segment. We conjecture that due to the similar texture in the background and foreground, inferring correct object regions from a static image can be harder than inferring them when the objects are moving. This may explain the significantly larger gap with SLATE in MOVi-Tex compared to the other datasets. Note that this gap exists even when evaluating STEVE with zero past frames (i.e. a static image), indicating that training on temporal data helps STEVE infer segments on static images.

## 5.2 Unsupervised Video Segmentation

In this section, we evaluate how well our predicted segmentation tracks the ground truth segmentation in both space and time without switching identities. For this, we compute *Video FG-ARI* which considers the full trajectory of a segment over the video length as one cluster. In line with the previous

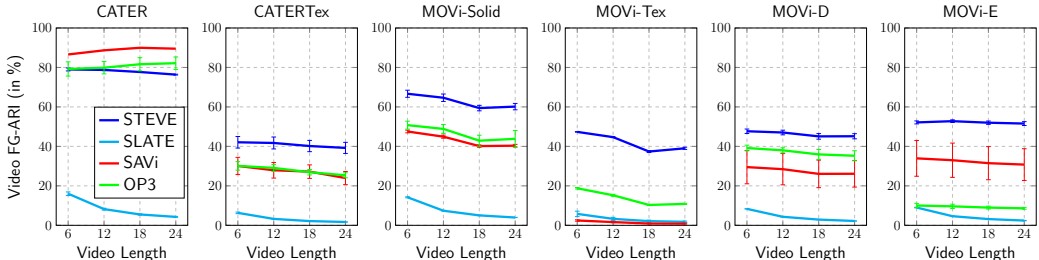

Figure 3: **Unsupervised Video Segmentation.** On the $y$-axis, we plot Video FG-ARI and compare our performance with the baselines. Along $x$-axis, we show how the performance is affected by the length of the video at test time.

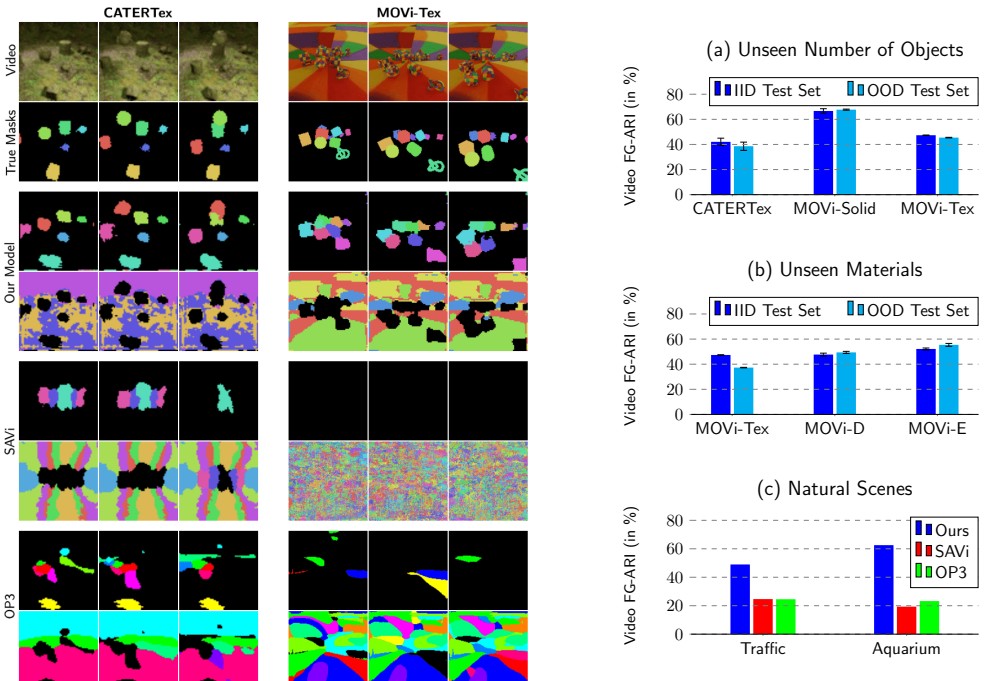

Figure 4: **Left: Visualization of Unsupervised Video Segmentation in CATERTex and MOVi-Tex.** We show the video frames and their true masks followed by foreground and background segmentation of each model. We separated the visualization of the foreground segments to make the result easier to interpret by applying a threshold on the segment area. See the supplementary material and the project website `https://sites.google.com/view/slot-transformer-for-videos` for more visualizations. **Right: Unsupervised Video Segmentation in OOD and Natural Datasets.** We plot the Video FG-ARI. In (a) and (b), we compare the performance of our model on IID and OOD test sets. In (c), we evaluate our performance in unsupervised video segmentation on natural scenes.

works Locatello et al. (2020); Kabra et al. (2021); Kipf et al. (2021), we consider only the foreground segments of the ground truth.

**Video Segmentation Performance.** In Figure 3, we see that in 5 out of 6 datasets, all of which are textured, STEVE significantly outperforms the baselines. In CATER, the baselines are able to perform comparably or slightly better than ours. As with the segmentation result of Section 5.1, this is not surprising given that CATER is visually simple enough for the baselines to handle well. We also observe the one major drawback of SLATE in these results i.e. SLATE cannot provide aligned slot representations for a video. This is because SLATE can only be applied to individual frames and the slots of each frame would be randomly permuted. We also show how the performance is affected by the length of the video. The models were trained on 3-length videos (6-length for CATER) while

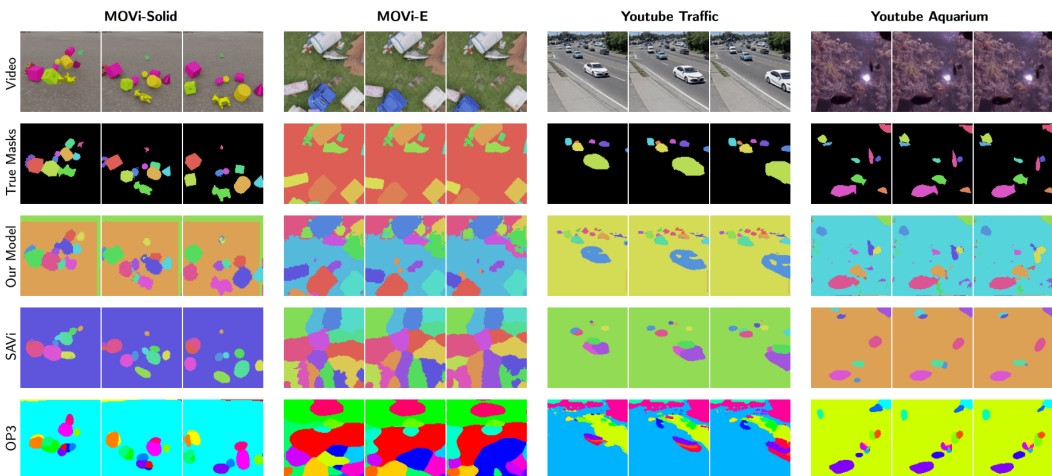

Figure 5: **Visualization of Unsupervised Video Segmentation in MOVi-Solid, MOVi-E, and Natural Scenes.** The rows (top to bottom) show the input video, the true segmentation followed by the predicted segments of the models. See the supplementary material and the project website `https://sites.google.com/view/slot-transformer-for-videos` for more visualizations.

we evaluate on up to 24-length videos. As the likelihood of switching identities should increase with the video length, all models show some downward slope. Going from video length 6 to 24 in CATERTex, MOVi-D, and MOVi-E, our model deteriorates noticeably less than the baselines.

**Out-of-Distribution Generalization.** In Figure 4 (a) and (b), we show how well our trained model can generalize at test time to out-of-distribution videos which have more number of objects and unseen objects and materials. We see that our model generalizes well and retains a similar performance as that on the IID test set. One exception is MOVi-Tex in which our model generalizes well to more objects but suffers on novel textures. Due to the modular structure of the recurrent encoder, each slot binds to input objects quite independently. We conjecture that this modularity is responsible for good generalization to more objects. However, for new materials, generalization depends on how well the backbone CNN can generalize. As the training set of MOVi-Tex was generated using a library of 240 textures, it would not be surprising if this CNN provides poor features of unseen textures. This, we believe, is affecting the performance of our model.

**Natural Scenes.** In Figure 4 (c), we evaluate video segmentation on two natural datasets taken from real-world videos: Traffic and Aquarium. We see that our model performs significantly better than the baselines, approximately doubling the FG-ARI in the Traffic dataset and tripling the FG-ARI in the Aquarium dataset. Noteworthy is the visual complexity of the Aquarium dataset in which the color and texture of fish are strongly camouflaged against the background and seeing them requires careful inspection even for a human (see Figure 5). Despite this, we note that our model is able to deal with this challenge effectively.

## 5.3 Analysis

**Diagnosing Slot Representation in STEVE.** In previous sections, we focused on segmentation quality and used input attention masks of STEVE. However, input attention masks do not provide a direct indication of the information content of slots. We would like to diagnose how well the slots represent the object-centric structure of the scene. For this, we take the slots from the pre-trained encoder of STEVE and, without propagating the gradient to the encoder, we train a mixture-based decoder to reconstruct the observation from the slots. We call this model *STEVE-Diagnostic*. In Figure 6 (a), we compare the performance of unsupervised video segmentation with the baselines using the decoding masks of STEVE-Diagnostic. The decoder of STEVE-diagnostic is made to be exactly the same as that of SAVi for a fair comparison. We find that STEVE-Diagnostic performs better than the baselines in all textured datasets, in line with our earlier results in Figures 2 and 3. This shows that the slots in STEVE represent the object-centric structure better than the baselines. In

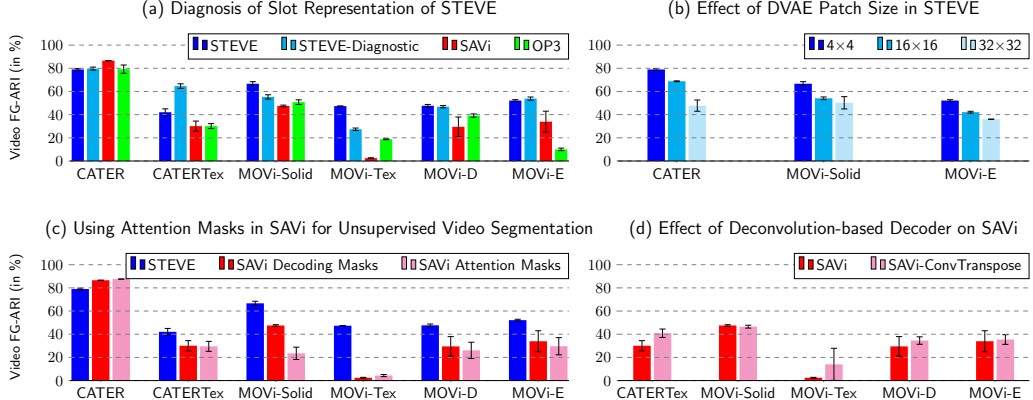

Figure 6: **Analysis.** Based on the analyses described in Section 5.3 we evaluate various aspects of the models. In all plots, we report Video FG-ARI computed on videos of length 6.

Table 9 in Appendix C, we further analyze the slot contents by training a linear regressor to predict the object positions and find that STEVE performs favorably relative to the baselines.

**Effect of Granularity of dVAE on STEVE.** In this section, we analyze the effect of granularity of discrete VAE on the segmentation performance of STEVE. In the default configuration of our model, the discrete VAE divides the image uniformly into patches of size $4 \times 4$ and represents each patch with one discrete token. That is, an image of size $128 \times 128$ would be represented by a sequence of length 1024. In this analysis, we test the effect on segmentation performance if dVAE represents the image using larger patches i.e. $16 \times 16$ and $32 \times 32$. In our results shown in Figure 6, we note that using larger patches results in a worsening of the performance. This suggests a possible scaling law that smaller patches lead to better performance. However, due to the prohibitive memory needs of the transformer on longer sequences, we were not able to experiment with smaller patches.

**Effect of Transformer Capacity on STEVE.** In Fig. 7 in the appendix, we analyze whether our segmentation performance is sensitive to the capacity of the decoder. For this, we test the effect of increasing the number of layers in the transformer decoder. We find that our segmentation performance is unaffected by this change, showing that our model is robust.

**Comparison between Decoding Masks and Input Attention Masks in SAVi.** In previous sections on unsupervised image and video decomposition, we used input attention masks of STEVE while for the baselines, we used the decoding masks. We would like to test how different are the performances of input attention masks and decoding masks. While OP3 only provides decoding masks, fortunately, SAVi provides both input masks and decoding masks which we compare in Figure 6 (c). We find that input attention masks tend to be worse than decoding masks of SAVi. Therefore, the attention masks of STEVE outperform both the input and decoding masks of the baseline SAVi.

**Mixture-based decoding without Spatial Broadcast.** Mixture-based decoding requires a CNN to decode the object image and the mask from each slot. To implement this CNN, the baselines prescribe using a Spatial Broadcast decoder, a type of CNN decoder that limits the expressiveness of the decoder to facilitate the discovery of objects. We would like to test if a more flexible CNN implemented completely using transposed convolutions would be enough to improve the performance or not. In Figure 6 (d), our results on the 5 textured datasets show that this change leads to a similar or slightly higher performance than the original decoder. However, the model still fails by splitting the image into fixed patches similarly to before and still performs significantly worse than our model. This shows that simply using a more flexible CNN with mixture-based decoding is not enough and the expressiveness of an auto-regressive transformer is important for achieving significant gains.

**Computational Requirements of STEVE.** In Table 6 in the Appendix, we empirically compare the computational requirements of STEVE and SAVi. We assess by how much the computational requirement change in going from the mixture-based to the transformer-based approach. We find that in videos with image size $64 \times 64$, STEVE requires similar resources as SAVi and about twice for image size $128 \times 128$. Given the quadratic memory complexity of transformers, the larger memory

demand of STEVE is not surprising. However, one should also consider that the mixture-based baselines fail almost completely on datasets like CATERTex, MOVi-Tex, MOVi-D, and MOVi-E, unlike ours. We also provide a big-$O$ analysis of the memory costs in Appendix B.1.

**Effect of Camera Motion.** Our segmentation results on MOVi-D and MOVi-E in Figures 2 and 3 provide insight into the effect of camera motion as the MOVi-E dataset introduces a moving camera to MOVi-D. We observe that camera motion in MOVi-E leads to a slight improvement in performance over MOVi-D. We conjecture that this is because a moving camera causes relative motion between objects and can help better in bringing out patterns that tend to remain stable under motion.

**Static Objects.** We also note that our model is effective even when static objects are present in the scene such as in CATER, CATERTex, MOVi-D, and MOVi-E. In methods that rely purely on optical flow for discovering objects can suffer significantly with static objects. For MOVi-D and MOVi-E, Greff et al. (2022) report FG-ARI of 19.4 and 2.7 using SAVi trained with optical flow supervision. Compared to this, our model achieves a significantly better 47.67 and 52.15 on these datasets, respectively, despite the static objects and without using any supervision whatsoever.

## 6 Conclusion

In this paper, we study the effect of a powerful transformer-based slot decoder for unsupervised object-centric video learning. For this, we propose a simple model, STEVE, and demonstrate that it can deal with various complex and naturalistic videos without any supervision by significantly outperforming previous state-of-the-art baseline models. In experiments, we also find that although the slot-transformer decoder alone is an important component, for videos it is essential to learn also from temporal information for dealing with complex videos. The results of this paper and the SLATE paper jointly suggest using a powerful reconstruction decoder for unsupervised object-centric learning which is contrary to the traditional view advocating the use of weak mixture decoders. This raises a question to the community: *is accurate reconstruction from slots all we need to capture the objectness?* In the future, it seems worth investigating further in this direction. Also, because we choose to investigate a minimal architecture in this paper, it would be interesting to see what gains we could obtain additionally by exploring more advanced architectures. Lastly, it would be interesting to investigate applying this method to large-scale datasets. We further elaborate on these in Appendix F.

## Acknowledgments and Disclosure of Funding

This work is supported by Brain Pool Plus (BP+) Program (No. 2021H1D3A2A03103645) and Young Researcher Program (No. 2022R1C1C1009443) through the National Research Foundation of Korea (NRF) funded by the Ministry of Science and ICT.

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
