# OpenReview forum: "Simple Unsupervised Object-Centric Learning for Complex and Naturalistic Videos"
_NeurIPS.cc/2022/Conference — NeurIPS 2022 Accept_

### Official Review · Reviewer_84xq · 2022-06-21

**Rating:** 8
**Confidence:** 4
**Soundness:** 4 excellent
**Presentation:** 4 excellent
**Contribution:** 4 excellent

**Summary:**

-- Post rebuttal update ---
The rebuttal addresses my concerns with new experiments and a new limitations section in the manuscript. I continue to support the acceptance of this paper. Rating: 8.

-- End of post rebuttal update---

This paper proposes a model, called STEVE, for object centric learning for video data. STEVE is a combination of a recurrent slot encoder (like SAVi) and the SLATE slot-transformer decoder. The combination works surprisingly well on relatively complex synthetic data and even in some real world video datasets. STEVE demonstrates better object centric decomposition when compared against baselines on the FG-ARI metric. It also generalizes OOD to unseen number of objects and unseen materials.

**Questions:**

No major limitations and questions. The paper is a solid submission. I have discussed questions and concerns in the previous section itself.

**Limitations:**

Societal impact has been discussed.

Limitations can be elaborated further. For instance, the memory complexity of SLATE has been discussed in the paper; but the time complexity of sampling an image has not been discussed. Since transformer decoders need auto-regressive token-by-token generation they tend to be very slow during inference (not a problem during training). Maybe this becomes a big issue when running this at high resolution (512x512 or 1080p)?

**Strengths And Weaknesses:**

### Strengths

- The proposed method is simply yet very effectively. STEVE does not use extra annotations in the first frame, nor does it need optical flow or other signal in its loss function. Purely using dVAE token reconstruction as the loss function (indireclty just pixel reconstruction), STEVE is able to decompose textured videos, handle static objects (MOVi-D dataset), and moving camera (MOVi-E dataset).

- The method has been tested on a wide variety of datasets. This gives more weight to the claims. The comparisons show that the SLATE decoder is significantly better than a spatial broadcast decoder in both per frame and video metrics. The paper thus also provides insights about the SLATE decoder. SLATE itself is very powerful for object centric learning.

- The analysis covers obvious concerns such as using encoder attention maps as opposed to decoder attention maps, and relevant ablations  such as dVAE granularity are included.

### Weaknesses
* Minor issues:
  - Missing legend in figure 3(a), please note how many objects were used in the in-distribution and OOD settings.

  - If possible include STEVE in figure 5(a). What's the performance of STEVE relative to STEVE-Diagnostic?

  - I think you mean "convolution transpose" not "Deconvolution" in page 9. Please describe the exact architecture of this convolution transpose network. The spatial broadcast decoder also does some convolution transpose operations. How is this decoder different?

  - An explicit big-O analysis would be great in the "computational requirements" paragraph.

* A model figure is missing. I have read SAVi paper so I understand this method but people new to STEVE would likely benefit from a nice figure.

* FG-ARI ignores background pixels. This does not penalize leaking of object segments into the background. Relying only on this metric is dangerous. Is it possible to report readout based metrics in these datasets? For instance the slot attention paper reported property prediction for CLEVR.

* It is still not clear why the transformer decoder, which mixes information from all the slots in the decoding step unlike the mixture decoder which is applied to each slot independently, still leads to object decompositions? Does one need to control transformer capacity to make this work? What is the most sensitive hyper-parameter here?

---

> ### Author Response · Authors · 2022-08-02
> **Response to Reviewer 84xq**
>
> We are happy to hear your positive comments and the recommendation! Also, thank you for the very helpful suggestions for improving our paper.
>
> > Not clear why the transformer decoder still leads to object decompositions. Is it sensitive to transformer capacity? What is the most sensitive hyper-parameter here?
>
> Based on our earlier experiments, the model did not seem very sensitive to the transformer capacity. To test this more explicitly, we ran an experiment on the MOVi-E dataset by increasing the number of decoder layers of the transformer decoder from 8 to 12 and we found the performances to be comparable in both these settings. For an 8-layer transformer, the model achieves an FG-ARI of **52.15 $\pm$ 0.76** while for a 12-layer transformer, the model achieves a quite similar FG-ARI of **52.95 $\pm$ 0.17**. Thank you for this great idea for showing further analysis of the model! We will also add this experiment for more configurations and other datasets in the final version.
>
> To our knowledge, the learning rates seemed to have the most effect on performance. But we do not think that this hyperparameter was too sensitive or hard to discover. It is also worth noting that for a given image size, we did not change any hyperparameters across datasets in our experiments. This provides some evidence that hyperparameters are reasonably robust and do not need to be tuned for each individual dataset.
>
> We agree that why the transformer decoder works is an interesting question requiring a thorough future investigation. Our current conjecture is as follows. We think that the major driver of object emergence is still the slot attention layer. Even if the transformer mixes information from all the slots in the decoding step, the slots still provide a bottleneck where they should compete to obtain the best spatial areas to attend to. The transformer decoder therefore should find a way to best use these slots by helping each slot find the best attention area. We think that the transformer decoder can actually do this very well by providing a better learning signal than the mixture decoder because of its expressiveness (e.g. the ability to render complex textures) and the ability to model relationships or interactions between slots and the already generated dVAE codes. It turned out that such an effective area for a slot to attend actually corresponds to an object. In contrast, the mixture decoders are usually weak and thus find it difficult to model complex textures or relationships between slots. Therefore, a poor learning signal is propagated to the slot attention layer.
>
> > What's the performance of STEVE relative to STEVE-Diagnostic?
>
> Thank you! We added the bars for STEVE in the same plot as STEVE-Diagnostic to ease comparison.
>
> We note that the decoding masks of STEVE-diagnostic generally seem to have a better performance than the attention masks of STEVE. This is because, unlike the attention masks, the decoding masks must try to cover the object area more fully. In this regard, the case of CATERTex is interesting where the FG-ARI of decoding masks of STEVE-diagnostic is about 50% higher than the attention masks.
>
> However, this technique of using STEVE-diagnostic for studying slot contents also has limitations. This is because it used a mixture-based decoder that is inherently weak. Thus in complex scenes, even if the slot contents are good, the decoder would not be able to properly render the visually complex objects and their decoding masks may get degraded. In some datasets like MOVi-Tex, we believe this effect is responsible for a drop in FG-ARI of STEVE-diagnostic relative to STEVE.
>
> > FG-ARI can leak background area into object slots. Discuss or experiment with property prediction using the slots.
>
> Thank you for the suggestion! We were not able to run this during the rebuttal period, but yes, we should be able to predict object color and shape on the CATER dataset using a similar matching procedure as the slot attention paper. We will consider this for the final version of the paper.
>
> > Auto-regressive token-by-token generation can be slow. Limitations can be elaborated further.
>
> Thank you, this is a great point! However, in the current work, we do not focus on the generation task. That being said, we also totally agree with you. We have added a discussion of limitations and also included this one in the revised version.
>
> > Readers would benefit from a nice figure. An explicit big-O analysis of computational requirements would be great.
>
> Thank you for these great suggestions! We will add them in the final version.
>
> > Provide how many objects were used in the in-distribution and out-of-distribution settings.
>
> Due to space limitations, we provided this detail in the appendix. However, we will find a way to include it in the main paper.

---

> > ### Comment · Reviewer_84xq · 2022-08-05
> > **Reply**
> >
> > I thank the authors for answering my questions.
> >
> > That's a nice result that the transformer doesn't need extensive tuning and that a decomposition still emerges. The new limitations section looks good.
> >
> > The authors have promised quite a few changes. While it is entirely up to them as to which of these they prioritize in the final version, I'd like to suggest trying to complete the property prediction experiment. This will help future research compare with STEVE regardless of the ARI/FG-ARI/mIoU/NMI or other segmentation/clustering metric and regardless of which masks - encoder/decoder/something-entirely-new get generated (IMHO).
> >
> > I continue to support the acceptance of this paper with a rating of 8.

---

> > > ### Author Response · Authors · 2022-08-06
> > > **Response to Reviewer 84xq**
> > >
> > > Thank you, we sincerely appreciate your continued support for our paper! As you suggested, we will prioritize the property prediction experiment for the final version.

---

### Official Review · Reviewer_88Gq · 2022-06-29

**Rating:** 7
**Confidence:** 4
**Soundness:** 4 excellent
**Presentation:** 4 excellent
**Contribution:** 3 good

**Summary:**

The paper proposes a new method for unsupervised object discovery and tracking in videos called STEVE. This model combines an encoding backbone, a recurrent slot encoder and a transformer-based decoder to achieve impressive object discovery results on complex, textured multi-object datasets. Additionally, the model provides useful insights into its predecessor, SLATE, which worked on static images.


**Questions:**

Q.1) From Figure 1, it becomes apparent that the model improves in performance when a couple of initial frames have been processed. Does this effect still hold when objects move very fast in between frames? Is there some kind of maximum speed of objects between frames that the model can handle?

Q.2) SLATE's big selling point was to be able to generate new multi-object images. Since STEVE is using a very similar architecture, would it be possible to generate multi-object videos with this model as well?


**Limitations:**

Limitations are adequately addressed in the paper.


**Strengths And Weaknesses:**

# Strengths

S.1) The writing style is great, and the paper is a pleasure to read. It is easy to follow and contains a lot of useful insights. In particular, I enjoyed the summarizing statements at the end of many paragraphs in the experiments section.

S.2) The proposed method is comparatively simple, yet very effective. It achieves impressive object discovery performance, especially on textured multi-object datasets.

S.3) The paper provides extensive and insightful experiments across 8 datasets, 5 of which are introduced by the paper itself.


# Weaknesses

W.1) The description of the model in l.119-128 was somewhat unclear to me. Is the dVAE an entirely separate model from the slot-encoder, transformer-decoder architecture? How exactly do the dVAE and transformer decoder interact, and why? Additionally, since this is heavily based on the SLATE model, it would be nice to explicitly mention the architectural differences between the two models.

# Others
The references need some clean-up: there are several papers that appear twice in the list.

---

> ### Author Response · Authors · 2022-08-02
> **Response to Reviewer 88Gq**
>
> Thank you for your helpful feedback and the questions. We are greatly encouraged by your positive feedback of our work!
>
> > Discuss whether STEVE can be used to generate multi-object videos.
>
> This is a great question! The ability to compose and generate novel videos is one of the main future works of our model. Currently, when trying to generate the images for our visually complex datasets (e.g. MOVi-E), we find that the generations still have some room for improvement. However, our focus in the current work is on segmentation performance rather than image generation.
>
> At the same time, it is encouraging to see the high-quality generations of natural images from many recent models like DALL-E, Image-GPT, and Parti, suggesting that a future extension of STEVE should be able to achieve this ability.
>
> > It would be nice to explicitly mention the differences between the SLATE model and STEVE. How do dVAE and the transformer interact? Is the dVAE an entirely separate model from the slot-encoder and the transformer-decoder architecture?
>
> Thank you for pointing! We briefly address them below. We have added a more in-depth description in Appendix B of the revised version. We welcome any further feedback! We will also release our code.
>
> The key difference between STEVE and SLATE is that STEVE temporally extends SLATE by adopting the recurrent slot encoder. Because of this, STEVE can provide important temporal information such as the slot-object tracking information which SLATE cannot provide. In doing this, STEVE retains the same decoder architecture as that of SLATE.
>
> The dVAE and the transformer interact when the transformer decoder takes the dVAE tokens as input and learns to predict the next token. We learn to do this next-token prediction for all tokens in parallel by applying causal masking in the style of GPT. In further detail, the dVAE tokens given as input to the transformer are obtained via argmax on the logits output by the dVAE encoder. That is, we take the dVAE logits and first perform argmax. For this argmax index, we retrieve a learned embedding from a codebook. To these retrieved embeddings, we add learned positional encodings and the resulting embeddings are then given as input to the transformer decoder. Similar to SLATE, we do not let gradient be propagated across the argmax operation. The prediction targets for the transformer are the argmax indices of the next dVAE token which we learn to predict using a cross-entropy loss.
>
> Although the full model (the dVAE, the slot encoder, and the transformer) is trained jointly in a single training run, we do not let gradients of dVAE’s learning objective (mentioned as $\mathcal{L}_\text{dVAE}$ in the paper) be propagated to the remaining network. In this sense, dVAE may be said to be separate from the rest of the model.
>
> > Discuss or experiment on how object speed affects video decomposition.
>
> Very interesting question! This is also a great idea for further analysis of the model. To test this, we ran an experiment on MOVi-Solid since this dataset has a large number of moving objects. We train STEVE on videos in which every alternate frame is skipped. This makes a video twice as fast as the original.
>
> - FG-ARI on MOVi-Solid (original speed): 66.66 $\pm$ 1.80
> - FG-ARI on MOVi-Solid (2x speed): 61.90 $\pm$ 0.63
>
> By doing this, we note degradation in performance. We also note that the performance warm-up resulting from observing the first few frames also decreases in the case of 2x speed. We show the plot for this ablation (analogous to Fig 1) at this anonymized link [https://imgur.com/a/FQZjyv7](https://imgur.com/a/FQZjyv7). We conjecture that this degrading trend would continue as the object speeds are further increased. We will test this further and add the results to the final version. However, we also believe that this behavior is more dependent on the encoder and not on the decoder which is the main focus of this work.
>
> > Some references appear twice.
>
> Thank you for pointing this out! We have fixed them in the revised version.

---

> > ### Comment · Reviewer_88Gq · 2022-08-03
> > **Reply**
> >
> > Thank you for your insightful reply!
> >
> > First, I would like to mention that I'm rather surprised by WXnF's review, and highlight that I do not agree with their negative evaluation of this work. Overall, I still believe that this paper should be accepted.
> >
> > Besides this, I have one follow-up question regarding the differences between SLATE and STEVE: Given that they largely share the same architecture, would SLATE be able to discovery objects in comparatively complex, static images? I think it would be interesting to add a discussion on this in the paper, as it could further clarify the relative contribution of STEVE. Additionally, I would recommend making the key difference between STEVE and SLATE (i.e. the recurrent slot encoder) explicit in the main paper, rather than the Appendix only.

---

> > > ### Author Response · Authors · 2022-08-06
> > > **Response to Reviewer 88Gq**
> > >
> > > > I would like to mention that I'm rather surprised by WXnF's review, and highlight that I do not agree with their negative evaluation of this work. Overall, I still believe that this paper should be accepted.
> > > >
> > >
> > > First of all, thank you for this! We really appreciate your support for our work. We will try our best to help the reviewer WXnF see the value of our paper.
> > >
> > > > I have one follow-up question regarding the differences between SLATE and STEVE: Given that they largely share the same architecture, would SLATE be able to discover objects in comparatively complex, static images? I think it would be interesting to add a discussion on this in the paper, as it could further clarify the relative contribution of STEVE.
> > > >
> > >
> > > Very good point. In the last paragraph of the Introduction, we actually discuss about this to some extent. We have copied it below. We welcome any further specific questions in this regard. Please let us know. We would be happy to answer!
> > >
> > > *“We ﬁnd that SLATE can deal with complex naturalistic images reasonably well despite not using temporal information. Although as a static model, SLATE has a fundamental limitation that it cannot take advantage of temporal information, e.g., maintaining consistent object identity across time, if we only look at its frame-level segmentation ability, it already performs better than the previous state-of-the-art temporal model that applies slot attention to videos (Kipf et al., 2021). Crucially, we also ﬁnd that SLATE alone is not enough since our model, STEVE, shows much better frame segmentation than SLATE, while also providing consistent tracking of slots in videos, which SLATE cannot do. In particular, we ﬁnd that on some complex videos such as MOVi-Tex, SLATE fails signiﬁcantly, suggesting that learning jointly from both temporal information and the powerful SLATE decoder is signiﬁcant and essential to realize the full potential of object-centric representation learning in videos.”*
> > >
> > > We also discuss our experimental results about these in Sections 5.1 and 5.2:
> > >
> > > *“Benefit of Temporal Learning. We also analyze whether training STEVE on videos has any benefit in image segmentation compared to SLATE. As SLATE is applicable only for static images, we train and test it on one-frame `videos' of our datasets. Comparing STEVE with SLATE, we note that STEVE performs consistently better in all datasets. Noteworthy is the gap in MOVi-Tex which is especially large. This suggests that training on temporal data i.e. videos can be helpful for learning to segment. We conjecture that due to the similar texture in the background and foreground, inferring correct object regions from a static image can be harder than inferring them when the objects are moving. This may explain the significantly larger gap with SLATE in MOVi-Tex compared to the other datasets. Note that this gap exists even when evaluating STEVE with zero past frames (i.e. a static image), indicating that training on temporal data helps STEVE infer segments on static images."*
> > >
> > > *“We also observe the one major drawback of SLATE in these results i.e. SLATE cannot provide aligned slot representations for a video. This is because SLATE can only be applied to individual frames and the slots of each frame would be randomly permuted.”*
> > >
> > > > Additionally, I would recommend making the key difference between STEVE and SLATE (i.e. the recurrent slot encoder) explicit in the main paper, rather than the Appendix only.
> > > >
> > >
> > > Thank you for this suggestion. We will do so in our final version.

---

### Official Review · Reviewer_kCr8 · 2022-07-11

**Rating:** 5
**Confidence:** 4
**Soundness:** 3 good
**Presentation:** 2 fair
**Contribution:** 3 good

**Summary:**

Learning the compositional structure (e.g. scene graph) in dynamic visual scenes without labels (e.g. object masks) has proven challenging.  Slot-based models leveraging motion cues have recently made progress in learning to represent, segment, and track objects without direct supervision. However, these methods only work on synthetic data and fails on complex real-world multi-object videos.  This paper proposes STEVE, an unsupervised model for object-centric learning in videos. The paper clams that it made a significant progress by demonstrating its effectiveness on various complex and naturalistic videos unprecedented. This is enabled by a simple architecture that uses a transformer-based image decoder conditioned on slots and the learning objective is simply to reconstruct the observation.  Experiment results on several videos show improvements compared to the previous state-of-the-art, SAVi, SLATE and OP3.

**Questions:**

The paper uses discrete VAE encoder to discretize each frame as a sequence of discrete tokens. The authors may want to try Vision-Transformer-based VQGAN (ViT-VQGAN) which has found successes in Parti (Scaling Autoregressive Models for Content-Rich Text-to-Image Generation, https://arxiv.org/abs/2206.10789).

The paper should provide more details on the two natural datasets. Furthermore, it should consider evaluating on existing benchmarks, e.g. Waymo Open Dataset.

The paper should provide more insights and characterize on when STEVE works and when it fails. Identify failure cases is important for future work.


**Limitations:**

The paper does not address its limitations. A section clearly articulating the technical limitations is important.

**Strengths And Weaknesses:**

Strengths

The paper proposes a a simple architecture that uses a transformer-based image decoder conditioned on slots and the learning objective is simply to reconstruct the observation. The transformer-based image decoder uses the one proposed in prior work, SLATE (Singh et al., 2022).

Weaknesses

Despite the claims on its effectiveness on complex and naturalistic videos, the datasets seem to be very limiting.  6 of them are procedurally generated datasets, CATER (Girdhar & Ramanan, 2020), CATERTex, MOVi-Solid, MOVi-Tex, MOVi-D, and MOVi-E (Greff et al., 2022). 2 natural datasets are only collected for this paper: Traffic and Aquarium. The paper does not provide details on the complexity and diversity of these datasets.

The paper seems to lack novelty as the design seems to be a straightforward combination of (1) a CNN-based image encoder, (2) a recurrent slot encoder that updates slots temporally with recurrent neural networks (RNNs), and (3) the slot-transformer decoder of SLATE.

---

> ### Author Response · Authors · 2022-08-02
> **Response to Reviewer kCr8**
>
> Thank you for your review and a positive recommendation for our work!
>
> We respond to the following three main weaknesses identified by you in the **Common Concerns** section at the beginning of the response.
>
> - Limited datasets
> - Model novelty
> - Identifying limitations
>
> We now respond to the remaining concerns below.
>
> > The authors may want to try ViT-VQGAN which has found success in Parti.
>
> This is a very nice suggestion! Scaling up this model is one of the main future works. In this regard, we totally agree that this suggestion is worth trying.
>
> > Details on complexity and diversity of natural datasets.
>
> While we agree that the complexity and diversity of our Youtube datasets are not as high as some established natural video benchmarks such as the suggested Waymo Open Dataset, **it still has enough complexity and diversity that makes the prior methods, in this line of research, suffer severely unlike our model.**
>
> We now provide some more details of these datasets. While we already provided some details of these datasets in Appendix A, we have also added the following descriptions about the diversity in our updated version.
>
> The Youtube-Traffic dataset was generated from a video feed collected by a traffic camera that watches a specific intersection. The camera does not move during the video. The lighting conditions change slightly over time and this has a slight effect on the background and the overall scene from one clip to another. The vehicles appear to show diversity in size, color, and type.
>
> The Youtube-Aquarium dataset was generated from a video collected by a static camera mounted inside a fish exhibit. The background, lighting conditions, and other characteristics remain stationary throughout the video. The size, color, and types of fish show some diversity.
>
> For both datasets, a 6-hour long stream was collected, center-cropped, resized, and divided into clips to obtain episodes for training and evaluation.

---

> > ### Comment · Reviewer_kCr8 · 2022-08-07
> > **Thank you**
> >
> > I appreciate the authors' effort addressing my comments. I think the paper has some merits. However, the major problems are still not adequately addressed.
> >
> > The paper only reports FG-ARI metric. Other related papers, e.g. SAVi also reports mean Intersection over Union (mIoU) which is a more direction metric on segmentation. I would encourage the authors' to provide mIOU metric in the final paper.
> >
> > I find the limitation section too general and not very actionable. (1) It is premature to talk about scaling the model to internet-scale datasets. As the problem of self-supervised object-centric learning is far from solved, it is better to stay focused on the next actionable steps, e.g. by identifying and tackling key failure cases of STEVE on current datasets and more complex and diverse real datasets than Traffic and Aquarium. (2) "adopting more advanced architectures", what advanced architectures are you referring to and why would you think they help? (3) "more computational efficiency is required in the future to handle images of higher resolution because the transformer decoder has a quadratic memory complexity and also because token-by-token generation can be slow. " This is very well-known. Please be more specific on actionable next steps. (4) "we found that the quality of image generation in naturalistic datasets has some room for further improvement", what needs to be improved?

---

> > > ### Comment · Reviewer_84xq · 2022-08-08
> > > **SAVi's mIoU metric**
> > >
> > > I haven't kept up with the whole conversation here but wanted to reply about SAVi's mIoU metric. They report this only in the conditional case (Semi supervised tracking in Computer vision terms). In this setting bounding boxes or segmentation masks are given in the first frame for the objects of interest. The model needs to predict segments for the remaining time steps. This metric is non-trivial to adapt to the fully unsupervised setting of STEVE. One would likely need to derive a new metric based on the [unsupervised DAVIS Jaccard-Mean metric](https://github.com/davisvideochallenge/davis2017-evaluation/blob/d34fdef71ce3cb24c1a167d860b707e575b3034c/davis2017/evaluation.py#L63).

---

> > > ### Author Response · Authors · 2022-08-10
> > > **Response to Reviewer kCr8**
> > >
> > > > The paper only reports FG-ARI metric. Other related papers, e.g. SAVi also reports mean Intersection over Union (mIoU) which is a more direction metric on segmentation. I would encourage the authors' to provide mIOU metric in the final paper.
> > > >
> > >
> > > As noted by reviewer 84xq, SAVi reports mIoU only for the semi-supervised setting and it is not straightforward to apply the same approach in a fully-unsupervised setting. In fully-unsupervised methods for videos, almost all previous works to our knowledge use FG-ARI. Nevertheless, we agree that mIoU also has its benefits, and thus we are discussing how we may incorporate it into the final version.
> > >
> > > > I find the limitation section too general and not very actionable. (1) It is premature to talk about scaling the model to internet-scale datasets. As the problem of self-supervised object-centric learning is far from solved, it is better to stay focused on the next actionable steps, e.g. by identifying and tackling key failure cases of STEVE on current datasets and more complex and diverse real datasets than Traffic and Aquarium. (2) "adopting more advanced architectures", what advanced architectures are you referring to and why would you think they help? (3) "more computational efficiency is required in the future to handle images of higher resolution because the transformer decoder has a quadratic memory complexity and also because token-by-token generation can be slow. " This is very well-known. Please be more specific on actionable next steps. (4) "we found that the quality of image generation in naturalistic datasets has some room for further improvement", what needs to be improved?
> > > >
> > >
> > > (1) Regarding scaling up the model and data, we feel that instead of doing it after object-centric learning is solved, it might be considered as a way to make it work. As we’ve seen with LLMs, just providing more data may open up a way to improve. Yes, we also agree that the large-scale experiment may not be easy to execute as an immediate next step.
> > >
> > > (2) Our current architecture is pretty minimal. So, it’s reasonable to think that various latest techniques might help obtain further improvement in our proposed framework. However, we do not know which architectural changes will be fruitful. It’s a future work that we *will* investigate.
> > >
> > > (3) Yes, we agree that it is a well-known issue but we also believe that it is still an important limitation in the context of scaling up the model in the future. Therefore, we find it worth mentioning.
> > >
> > > (4) We mentioned that the quality of image generation in naturalistic datasets needs to be improved because, in the current implementation, the visual quality of reconstructions is a bit blurry. Addressing this is also a future work and we believe there is promise in this direction since many recent auto-regressive transformer decoders have shown good results.
> > >
> > > Most importantly, we will release our code here (anonymous link): [https://github.com/video-slot-transformer/](https://github.com/video-slot-transformer/). This will allow the research community to explore the limitations more deeply.

---

### Official Review · Reviewer_WXnF · 2022-07-11

**Rating:** 3
**Confidence:** 5
**Soundness:** 1 poor
**Presentation:** 2 fair
**Contribution:** 1 poor

**Summary:**

This work attempts to learn improved slot-based object representations and video segmentations with the help of the SLATE/Image GPT [2] decoder. The model is evaluated on several datasets included ones generated by the authors.

**Questions:**

**Claims**:
1. Page 2, line 87: “The emergence of object-centric slots is strongly dependent on what decoder is used.” Could you please provide supporting evidence for this.
2. Page 3, line 92. “The reconstruction objective is taken to be the squared error between the input and the reconstructed image.” This is not always the case and I would strongly caution against this over-simplification. Typically you would use the mixture likelihood $p(x) = \sum_n m_n \odot p(x; \hat{x_n}, \sigma)$ rather than an MSE wrt to $\hat{x} = \sum_n m_n \odot \hat{x_n}$, which is a lossy point estimate that defeats the purpose of a mixture model.
3. Page 9, line 209: “Therefore, in all the comparisons, the decoding masks gave an advantage to the baselines but not to our model.” This claim is far from obvious. If SAVi’s decoder masks are better than its own attention masks, that is due to either the smoothing effect of the decoder or representation quality/information bottleneck in SAVi specifically. A model with a different loss/architecture/set of hyperparameters may behave entirely differently. For example: if we shrank the decoder size (number of layers/parameters) to make it underfit, the decoded segmentation performance will likely undershoot the attention mask performance. Hence, whether STEVE is at an advantage or disadvantage cannot be ascertained by looking at SAVi's performance alone.

**Methodology**:
1. Why not use an existing natural image/video dataset? It is impossible to judge the quality of Traffic and Aquarium especially given the segmentations were derived by manual annotation.
2. OP3 is not the best choice of baseline. Its performance is not meaningfully different from SAVi’s (except on MOVi-E where it suffers from not being able to handle camera movement). Its inclusion does not shed any light on STEVE’s performance. Could you consider at least an improved version of OP3 (e.g. PARTS [1])?
3. Figure 3: “We separated the visualization of the foreground segments to make the result easier to interpret by applying a threshold on the segment area.” This makes the use of the ARI-FG metric somewhat disingenuous. It is an acceptable measure so long as the discarded background pixels correspond to a single or separately modeled (e.g. in MONet or SPACE respectively) component. Can you reliably extract the background components from STEVE? Is your threshold fine-tuned per dataset? Do you use the same threshold across the baseline models?

**Results**:
1. Figures 3 and 4: In both CATERTex and MOVi-Tex, the model tends to split the textured background into multiple components. But it tends to produce a solid background on both Traffic and Aquarium data. Could you explain this difference in behavior, despite the obviously textured background in the natural datasets?
2. movi-d-002.gif: the falling object takes at least 4 different slots over time (green, blue, yellow, pink). Surely this is far from desirable? There’s no camera motion in this video. For human eyes, the moving object is trivially trackable.
3. catertex-001.gif: there’s a single slot (pink) for multiple object shadows, even when the objects have different dynamics. If the model took its cues from object dynamics (rather than colors), surely you would expect each object’s shadow to be associated with that object’s slot?

**Writing**:
1. There’s a lot of repetition between the abstract and introduction. Moreover, sentences like the following (Page 2, line 59) are superfluous in a technical submission to a research conference: “From these experiments, we discover important new knowledge to share with the community.”
3. Page 3, line 124: could you please clarify the dimensionality of each $z_{t, l}$? Are they one-hot vectors indexing into the discrete codebook? There’s other model details which would be useful to include in the main paper.
4. Figure 1: what are the error bars computed on?
5. Could you please cite [2] which is clearly the inspiration for the decoder architecture? It is not sufficient to cite SLATE alone.

**References**:

[1]  Zoran, D., Kabra, R., Lerchner, A., & Rezende, D. J. (2021). PARTS: Unsupervised segmentation with slots, attention and independence maximization. In _Proceedings of the IEEE/CVF International Conference on Computer Vision_ (pp. 10439-10447).

[2] Chen, M., Radford, A., Child, R., Wu, J., Jun, H., Luan, D. &amp; Sutskever, I.. (2020). Generative Pretraining From Pixels. _Proceedings of the 37th International Conference on Machine Learning<_, in _Proceedings of Machine Learning Research_ 119:1691-1703 Available from https://proceedings.mlr.press/v119/chen20s.html.


**Limitations:**

There was no attempt to explain the limitations of the model.

**Strengths And Weaknesses:**

The results have dubious merit, the claims are poorly supported, and the writing needs a lot of improvement. There is little innovation in experimental methodology. On the plus side, the paper conducts a wide evaluation over several datasets, but the central claim that STEVE works on naturalistic images is not convincing at all. It is also unclear whether STEVE is any different than SAVi with a SLATE/Image GPT [2] decoder.

---

> ### Author Response · Authors · 2022-08-02
> **Response to Reviewer WXnF (1/3)**
>
> > FG-ARI metric is disingenuous.
>
> The reviewer claims that computing FG-ARI requires manual thresholding. **This is not true**. We compute FG-ARI in the standard way in which it is done in the prior works e.g. IODINE, Slot Attention, SiMONe, SAVi. In line with these works, our work also does not use any thresholding. Therefore, the FG-ARI metric is not disingenuous.
>
> What we wrote in the caption of Figure 3 i.e. *"We separated the visualization of the foreground segments to make the result easier to interpret by applying a threshold on the segment area."* only applies to the qualitative visualization shown in Figure 3 (left) and is not applicable to any other part of the paper. In this visualization, note that this threshold does not affect the segmentation at all. It only determines which segments are visualized in the upper row while the rest are visualized in the lower row. This is not a problem because our focus is on understanding the segmentation and the background-foreground split is only to help see the segments with a less cluttered visualization.
>
> > There is little innovation in the experimental methodology.
>
> We do not claim to do innovation in the experimental methodology.
>
> > The writing needs a lot of improvement. There is repetition between the abstract and the introduction. Some sentences are superfluous such as “From these experiments, we discover important new knowledge to share with the community.”
>
>
> While there is always some room for improvement, we do not agree with these examples. First, it is natural for there to be some overlap between the abstract and the introduction. Second, we also do not see any issue with the sentence: “From these experiments, we discover important new knowledge to share with the community.” We feel it is simply a matter of style and the message we want to deliver from this sentence is quite clear. Here, “new knowledge to share with the community” is simply equal to our contributions. Any paper should have new knowledge to contribute to the community. We also believe that these findings are important in this specific line of research and we provide supporting arguments in the paragraph about why we believe so. Of course, it’s up to the readers to decide whether it is indeed important. But as far as writing is concerned, it should not be a problem for authors to reveal their own interpretation of the significance of their results. Other reviewers seem to be positive about what we interpret. As we said, we feel it is simply a matter of style. We believe that having a different style should not be considered a weakness.
>
> If there are other parts of the paper where the writing is unclear or could use improvement, we would be glad to take a look and revise, but we do not think it is fair to say “the writing needs a lot of improvement” with just these examples. In fact, other reviewers commented positively about the same writing by saying that our paper was “a pleasure to read” and our “writing style was great”.
>
> > Some failure cases.
> >
> > movi-d-002.gif: the falling object takes at least 4 different slots over time. Surely this is far from desirable? For human eyes, the moving object is trivially trackable.
> >
> > catertex-001.gif: A single slot models multiple objects’ shadows. If the model takes cues from object dynamics rather than colors, then it is expected that each shadow would be associated with the slot of the corresponding object.
>
>
> Yes, these are failure cases which could happen sometimes. We agree that a perfect model would not have these failure cases. However, we do not claim our model is perfect or achieves human-level performance. This is how the state of the art in this line of research currently works — which is far from human-level yet. Even if we can investigate some failure cases, our model should be judged based on the general and standard performance metric, e.g., as shown in Fig 1 and 2, rather than some specific failure cases. (This result can be interpreted as: our model has much fewer failure cases than the baselines.)
>
> > The claims are poorly supported.
>
> No, we do not agree. Our results computed on a wide range of naturalistic datasets clearly support our claim. Our quantitative performance as measured by the FG-ARI score in Figures 1 and 2 is clearly and significantly superior to the previous methods on textured and naturalistic scenes. Qualitatively, on many of these naturalistic datasets, previous methods fail by trivially splitting the image into fixed patches. The fact that our results support our claim is a view also shared by many other reviewers who noted that our method is “very effective”, our object discovery performance is “impressive” and our evaluation on a wide variety of datasets “gives more weight” to our claims.

---

> > ### Author Response · Authors · 2022-08-02
> > **Response to Reviewer WXnF (2/3)**
> >
> > > ‘Impossible’ to judge performance using manually annotated segments in Traffic and Aquarium datasets.
> >
> > No, we strongly do not agree on this.
> >
> > First, our major result is on the naturalistic MOVi series and the CATERTex datasets (Figures 1 and 2) which are already significantly more complex and naturalistic than the datasets of previous works in this line. As these are procedurally generated, they provide high-quality ground-truth segmentation labels for evaluation. On these datasets, our method already shows consistent and significant improvements and thus provides clear evidence of the benefit. Therefore, it is not “impossible” to judge the performance of our model.
> >
> > Second, when dealing with natural images, manual labeling is the usual way. Exact ground-truth labels are only available for simulated scenes like MOVi datasets. Therefore, we feel that it is unfair to say that it is “impossible” to judge the quality simply because it is manually labeled. What is important is not whether it is manually annotated or not, but the quality of the manual labels. We used a dedicated tool to help human segmentation as is usually done in natural datasets. We see the quality of these annotations is reasonable enough and comparable to other typical manual annotations used for natural images. We also shared the samples in the Appendix and we provide some more samples at this anonymized link [https://imgur.com/a/2RQ1iq9](https://imgur.com/a/2RQ1iq9). That being said, the claim that it is “impossible” to judge the performance on these two datasets is highly overstated.
> >
> > > Supporting evidence that the emergence of object-centric slots is strongly dependent on the decoder used.
> >
> > The supporting evidence can be found both in the literature as well as in our experiment results. In literature, Greff et al. (2019) show the effect of changing CNN decoder architecture on object discovery in the context of their model IODINE. Recently, SLATE (Singh et al. 2022) has also shown the effect of changing the decoder from CNN to an image transformer on object discovery. In our own experiments, STEVE, whose key difference with SAVi is the decoder, has a very different segmentation performance relative to SAVi (Figures 1 and 2).
> >
> > > The distinction between the MSE objective and the mixture log-likelihood objective is oversimplified.
> >
> > Thank you for pointing. What we wrote describes just one variant of the mixture-based decoder. We will clarify this in the writing. We focused on this variant because our main baseline Slot-Attention Video (SAVi) and other current state-of-the-art models in unsupervised object discovery such as Slot Attention use this MSE-based approach. Yes, we acknowledge that another common variant in the literature uses a VAE framework with a mixture log-likelihood objective. We already cite these works in our related work. It is also worth noting that our other baseline, OP3, uses a mixture log-likelihood objective rather than MSE. However despite this distinction, OP3 also fails quite similarly to SAVi on textured and naturalistic scenes in our experiments.
> >
> > > Whether STEVE is at an advantage or disadvantage cannot be ascertained by looking at SAVi's performance alone.
> >
> > Thank you, you are right that the performance of SAVi cannot be indicative of the performance of STEVE. We have clarified and fixed this in the writing. What we intended to highlight in the analysis of Fig 5(c) is that regardless of whether we use SAVi’s attention masks or its decoding masks, both still perform significantly worse than our model.
> >
> > > The inclusion of OP3 does not shed any light on STEVE's performance. Experiment with the baseline model PARTS.
> >
> > No, we do not agree on this. Although the two methods share some similarities, they still have quite different architectures: OP3 adopts a VAE framework while SAVi does not. Also, the slot encoder of OP3 has a very different architecture than SAVi. Given these differences, it is not obvious beforehand how they will perform relative to each other. Therefore, seeing their similar performance and their failure to deal with naturalistic videos is still meaningful and adds weight to our claim of STEVE's superiority.
> >
> > Thank you for the suggestion regarding the PARTS model. Unfortunately, the authors do not provide the code or detailed enough information for reproducibility. Any other stable implementations of this model, to our knowledge, are also not available.

---

> > > ### Author Response · Authors · 2022-08-02
> > > **Response to Reviewer WXnF (3/3)**
> > >
> > > > The background segment is solid in Traffic and Aquarium datasets but not in CATERTex and MOVi-Tex. Why?
> > >
> > > Our Youtube datasets, although natural, are still simple. One of the simplifying characteristics they have is fixed background. This is why the model finds it much easier to represent the background as a single slot. In contrast, in CATERTex, MOVi-Tex, MOVi-D and MOVi-E, the background varies from scene to scene due to jitter in camera angle, lighting, and also reflections and shadows cast by the objects. This difference, we believe, is responsible for their different behaviors.
> > >
> > > > Dimensionality of $\mathbf{z}_{t,l}$
> > >
> > > Thank you. We had already provided the detail in the appendix. Due to the limitation of space, we could not include it in the main paper. $\mathbf{z}_{t,l}$ is a discrete random variable that has 4096 possible values. We represent it in our model as a one-hot vector of size 4096.
> > >
> > > > In Figure 1, what are the error bars computed over?
> > >
> > > The error bars are computed over 3 training seeds.
> > >
> > > > Cite Chen, Mark et al. “Generative Pretraining From Pixels.” *ICML* (2020).
> > >
> > > Thank you for pointing. Yes, we intended to cite it. It was mistakenly removed during editing. We have added it in the revised version.

---

> > ### Comment · Reviewer_WXnF · 2022-08-03
> > **Regarding FG-ARI**
> >
> > How would you discard background slots for downstream applications in the common case where you don't have a ground-truth background mask? This is a real problem when STEVE splits the background across multiple slots (see Figure 3, showing a CATERTex background split across 3 slots and a MOVi-Tex background split across 4).
> >
> > That is why I think FG-ARI is disingenuous here, as it is unclear to me whether STEVE can output (or permit the selection of) "foreground objects" alone.

---

> > > ### Author Response · Authors · 2022-08-06
> > > **Response to Reviewer WXnF**
> > >
> > > > How would you discard background slots for downstream applications in the common case where you don't have a ground-truth background mask? This is a real problem when STEVE splits the background across multiple slots (see Figure 3, showing a CATERTex background split across 3 slots and a MOVi-Tex background split across 4).
> > > >
> > > >
> > > > That is why I think FG-ARI is disingenuous here, as it is unclear to me whether STEVE can output (or permit the selection of) "foreground objects" alone.
> > >
> > > No, using FG-ARI as an evaluation metric does not imply discarding BG slots in downstream tasks. We do not claim that. We think it is most general to provide all the slots for the downstream task. It’s up to the downstream model to decide how to use these slots. We believe that an attention-based model should be able to learn to select only the relevant slots.
> > >
> > > Regarding splitting of background into multiple slots, there is no reason to believe that such background split is harmful. It is also unfair to say that the split is a unique problem of our method. As shown in Fig 3, all baselines split the background (again, we don’t believe split should be a bad thing). What’s important isn’t whether the split happens or not but how it splits. If the split follows the true underlying structure in the background, it should be even a better thing than putting the whole background into a single slot because the latter can prevent modeling relationships between background components or between some FG slots and some BG slots.
> > >
> > > Our qualitative results in Fig 3 and 4 show that our split of the background follows the semantics of the background more faithfully than other methods. In this regard, the result in Fig 4 on MOVi-E is interesting: the scene contains some dry leaves which are unlabeled in the ground-truth segmentation while our model is actually able to discover those dry leaves.
> > >
> > > Also, FG-ARI has been a standard method in most previous works in this line (such as IODINE, Slot Attention, SIMONe, and SAVi). There is no reason for it to become an issue only from our paper. In some previous works, there was not much background split. However, it is not because the methods were more robust to splitting than ours but simply because the scene was too easy (like CLEVR or CATER) for the split to happen.
> > >
> > > Lastly, we would kindly and respectfully like to ask the reviewer to consider words like “disingenuous” with more care. We believe it is possible for us to have a more constructive discussion without using such sensitive words decisively before having enough discussion.

---

### Author Response · Authors · 2022-08-02
**General Response (1/2)**

We thank all the reviewers for their insightful feedback. We are encouraged that reviewer 84xq found our work a **solid submission** and our model to be **simple**, **effective,** and **working surprisingly well on relatively complex synthetic data and even in some real-world video datasets**. We are also encouraged that reviewer 88Gq found our **writing style great**, the paper **a pleasure to read**, our model **simple yet effective**, our object discovery **performance impressive,** and our **experiments extensive and insightful**.


## Common Concerns

> The model is a straightforward combination of known components.

As we describe in the paper, we would like to say that our model is simple and effective -- but it is hardly obvious or straightforward. Yes, the components like the image transformer and slot-based RNNs were known individually. But the fact that their simple combination can actually work much better than the previous more complex methods was not known and was not obvious before our paper -- in this sense, our result is also a bit surprising. From this perspective, we'd like to emphasize that this combination is "new", even if it's a simple combination, and the effectiveness of this simple combination was not known before, so the impact. Moreover, we believe that if one can achieve both performance improvement and reduce the complexity of a method, it is actually a desired thing, not a bad thing because of the simplicity. Indeed, due to this simplicity, we observe that our model works quite stably without very sensitive hyperparameter tuning.

Furthermore, as mentioned in the paper, we intentionally choose to stay in this minimal architecture. It is because this simple combination is the actual core and essence of the improved performance even if some may feel that it’s too simple nevertheless. We may obtain additional performance gain by adding some other components (probably by adding complexity) but we did not pursue that in order to deliver a clear lesson to the readers. As mentioned in the Conclusion, we think that on top of this work one may find a better and more novel architecture in the future. But, we believe that this study with the bare-bone architecture provides the most clear and honest message to the community.

> Datasets are limiting. Why not evaluate on an existing natural dataset e.g. Waymo Open Dataset?

Whether our datasets are limiting or not should be considered in the context of fully-unsupervised object-centric learning. In settings where some form of supervision is available, natural datasets are quite common. However, our setting is different from this. In our setting, the model must learn by observing only the RGB video and no other input or supervision is available.

In this setting, only very simple datasets like CLEVR or CATER could be handled effectively before our work. In particular, achieving good performance on the MOVi datasets should be considered a major achievement in this line of research. These datasets were created to raise the level to more complex and naturalistic videos in the community. Most importantly **no previous works (including unsupervised SAVi) have been shown to work on these datasets in a fully unsupervised way**. Therefore, from the perspective of comparing apples to apples (in this line of research), it is appropriate to see this benchmark as not being limited but rather trying to ‘expand’ the reach to more complex, naturalistic, and significantly challenging settings. We make progress in this direction by showing good performance on these for the first time.

Furthermore, we introduce the CATER-Tex and MOVi-Tex datasets as well as two real-world YouTube datasets. Having complex textures on both objects and the background is very challenging and on the MOVi-Tex dataset, our method shows a very big performance gap compared to SAVi. This shows how difficult this dataset is for previous methods. Although more natural and complex datasets will always be nice to have, we hope that the reviewers would see where the previous methods were standing before our work in this line of research.

Waymo Open Dataset is a great suggestion! We find this to be a very interesting future work where we improve the model and try even more complex natural videos.

---

> ### Author Response · Authors · 2022-08-02
> **General Response (2/2)**
>
> > A discussion of limitations
>
> Thank you for pointing this out. We fully agree that we need to discuss more about the limitations of this work. Below, we describe the main limitations and we have also updated them in our revised version.
>
> First, there is some opportunity to improve the performance further within the proposed framework by adopting more advanced architectures than the minimal architecture we choose in this work. We would say that this is more of a future work than a limitation of this framework. Second, more computational efficiency is required in the future to handle images of higher resolution because the transformer decoder has a quadratic memory complexity and also because token-by-token generation can be slow. Third, the potential to scale the model to internet-scale datasets is discussed in this work but not investigated. Our current datasets typically contain about 200K video frames with an image resolution of 128. For scaling this up, we may also need to develop a new model in this framework. Fourth, although our focus in this paper is not on synthetic image generation, we found that the quality of image generation in naturalistic datasets has some room for further improvement. We believe that this could not only provide the ability to synthesize images and videos but also improve object decomposition as well. Fifth, even if our model outperforms previous works in general, we also observe some failure modes in MOVi-E such as splitting of large objects or merging of very small objects. To resolve this, we believe that it’s worth investigating on a much larger dataset with a bigger model or with a more sophisticated architecture. Lastly, the CNN backbone of our slot encoder is rather lightweight, having just 4-layers and a feature-size 64. For scaling this model to more complex natural images, it is likely that this lightweight CNN will not be enough.
>
> ## List of Revisions
> Incorporating the feedback of the reviewers, we have also made the following revisions to the submission.
> 1. Fixed the duplicate citations.
> 2. Fixed the legend of Figure 3(a).
> 3. Added STEVE in Figure 5(a) to show the relative performance of STEVE and STEVE-diagnostic.
> 4. Corrected the writing by replacing ‘deconvolutions’ with ‘transposed convolutions’.
> 5. Added the details of the CNN decoder used in the ablation of SAVi implemented only using transposed convolutions in Appendix B.
> 6. Added the previously missed citation: Chen, Mark et al. “Generative Pretraining From Pixels.” *ICML* (2020).
> 7. Added a discussion of limitations in Appendix F.
> 8. Added more details of the architecture focusing on the interaction of the transformer decoder and the dVAE in our model. These are added in Appendix B.
> 9. Made a clarifying fix to the writing regarding the ablation of Fig 5(c).

---

### Meta-Review · Area_Chair_5v56 · 2022-08-27

**Recommendation:** Accept
**Confidence:** Certain

**Metareview:**

The paper proposes a method for unsupervised learning of objects from videos. In particular, the proposed approach combines two existing ideas - a recurrent slot-based architecture (like SAVi) and an autoregressive image decoder (like SLATE). The methods is thoroughly evaluated and shown to outperform the relevant baselines.

After considering the authors' feedback and extensive discussions, the reviewers' opinions of the paper are still mixed: two strongly positive, one neutral leaning positive, and one strongly negative. The key strengths and weaknesses that were pointed out are as follows:
Strengths:
1. Simple and efficient model
2. Convincing experimental results on 8 datasets, including 2 real-world
3. Insightful analysis and ablation studies

Weaknesses:
1. Lack of novelty
2. No experiments on more complex real-world datasets
3. FG-ARI is not a perfect evaluation metric
4. Lack of insight from the experiments

Note that some of the mentioned weaknesses are in conflict with strengths - for instance S1 with W1, S2 with W2, S2/3 with W3/4. Taking all of the above into account, I believe the paper presents a simple yet efficient combination of two existing methods and evaluated it thoroughly, in line with (and actually somewhat above) what is commonly done for unsupervised object learning papers - leading to good performance and interesting insights. I believe the simplicity of the method is valuable and should not be critiqued as lack of innovation, and the provided experiments are already on more complex datasets than commonly used in the field.

Therefore at this point I recommend acceptance, but encourage the authors to take the reviewers' comments to heart (whenever possible) and adjust the paper accordingly.

**Award:**

No

---

### Decision · Program_Chairs · 2022-09-14

Accept